# Calibration of Target-Tracking MIMO Radar Sensors by Comparison with a Narrow-Beam CW Doppler Reference

**Seif Ben-Hassine *** , **Jean-Marie Lerat *** , **Jimmy Dubard, Pierre Betis and Dominique Renoux**

High Frequency Metrology Department, Laboratoire National de Métrologie et d'Essais LNE, 78197 Trappes, France; jimmy.dubard@lne.fr (J.D.); pierre.betis@lne.fr (P.B.); dominique.renoux@lne.fr (D.R.)

* Correspondence: seif.benhassine@lne.fr (S.B.-H.); jean-marie.lerat@lne.fr (J.-M.L.);
  Tel.: +33-130-691306 (S.B.-H. & J.-M.L.)

**Abstract:** This paper introduces a method for calibrating radar sensors in a multi-sensor cinemometer system using a reference cinemometer based on CW Doppler radar. The method involves synchronizing sensors, pairing data with reference measurements, and performing polynomial corrections. Tests conducted on various traffic sites demonstrate the accuracy and reliability of the calibration process. Results show low uncertainties compared to regulatory standards. Validation against a calibrated lidar system confirms accuracy. This method ensures precise speed measurements, surpassing regulatory requirements, and demonstrating practical applicability in real-world scenarios.

**Keywords:** cinemometer; calibration; uncertainty; speed measurements; MIMO; metrology; road traffic control

## 1. Introduction

Speed control for law enforcement application has evolved significantly with the development of modern MIMO radar sensors and lidar sensors. New features such as measurement on curves, simultaneous measurement of several targets in multiple lanes, and speed gradient measurement are now common, and their validation requires the development of new reference systems. Existing solutions are based on reference cinemometers using a pulsed Continuous-Wave (CW) radar with a center frequency of 24.125 GHz and a fixed incidence angle of 25° [1]. This type of solution provides reliable speed measurements but suffers from several limitations, such as limited multi-lane detection capability, installation restricted to straight-line configurations, and the ability to detect only a single target at a time. Other approaches found in the literature include piezoelectric sensors and inductive loops installed in the road [2]. These solutions offer good measurement accuracy but require a dedicated test circuit and do not allow for the relocation of the reference system to different sites. This makes it challenging to test cinemometers on single-lane roads, multi-lane traffic scenarios, or curved roads.

Finally, the reference system for law enforcement cinemometer validation can also be a speed measurement system integrated into a test vehicle. GPS-based solutions have been tested with various cinemometers in [3]. These solutions enable the verification of cinemometers, provided there is accurate synchronization between the device under test and the GPS-based reference speed sensor.

Consequently, a new metrological reference was developed at Laboratoire National de Métrologie et d'Essais (LNE) to provide a solution that adapts to these advancements and addresses the shortcomings of existing methods. This new reference allows on site testing of cinemometer systems with improved detection capabilities and new functionality

compared to the previous one which is based on a single CW radar. The latter is called HADER. The new LNE system uses a multi-sensor approach to combine the data provided by the various sensors and determine the reference speed value and the associated uncertainty. The measurement uncertainties at each point measured by each sensor must be determined to propagate these uncertainties into the reference speed measurement using a data fusion module. A calibration method is required to assess the uncertainty budget for the various sensors.

This paper presents a method for calibrating the radar subsystem used in the LNE system by comparison with the narrow-beam CW Doppler reference, HADER. This method includes:

- A method for placing and measuring the position of the sensors to be calibrated, the reference sensor and the traffic lane.
- A method for synchronizing the sensors in time.
- A method for pairing the data and interpolating the measured speed value.
- A method for performing polynomial corrections on reference sensor calibration data, with interpolation of the corrections according to the measured speed.

## 2. Description of the Sensors Under Calibration

LNE has designed a new reference multi-sensor cinemometer system [4] to provide an accurate vehicle speed measurement and reduce uncertainty levels. The system uses multiple sensors, including four radars and one lidar speed sensor, controlled by an acquisition and processing unit. A block diagram in Figure 1 illustrates the different components of the system. Time synchronization is achieved through the MasterClock GMR1000 system [5,6], which is designed to deliver a clock to the PC used as an acquisition and control unit via a serial interface and a PPS (Pulse Per Second) signal. The GMR1000 synchronizes with GPS/GNSS signals and functions as an NTP (Network Time Protocol) server, providing a highly accurate clock in the event of a GPS connection failure.

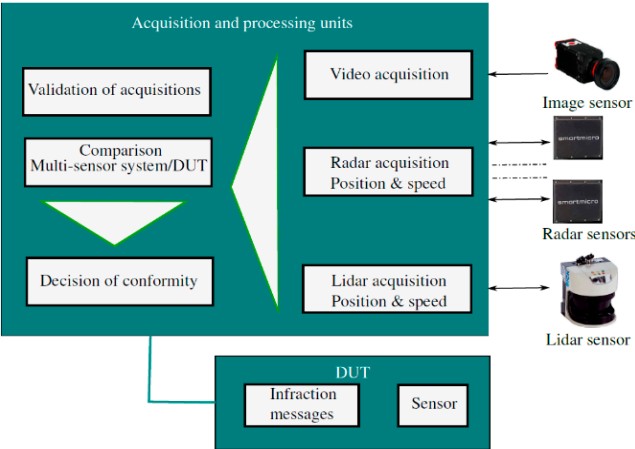

**Figure 1.** Block diagram of LNE cinemometer system.

This study focuses on the calibration of the individual speed values measured by radar sensors. The selected radar sensors are four FMCW MIMO (Frequency Modulated Continuous–Wave Multiple Input Multiple Output) components that integrate detection, ranging, and tracking functionality. The model selected is a SmartMicro UMRR-11 T132 [7], which provides a good detection range and resolution in positioning the target. This radar is designed for traffic management applications with multi-target tracking in multi-lane scenarios. The operating frequency range is between 76 GHz and 77 GHz. The sensor is based on 4D/HD+ technology [8], which allows for the separation of objects according to

their speed and distance from the sensor. When a vehicle enters the detection area of the radar, the sensor will initiate a tracking process that produces a list of target information, periodically updated, including speed, position, and type of vehicle at approximately 55 ms intervals. The algorithm employed for the tracking process is based on a Kalman filter [9,10], which estimates the position of the target from the measured range and angular position. The radar sensors are controlled by using a configuration file that employs B-Spline parameters, enabling accurate traffic lane approximation [11,12]. These parameters are the control points of the B-Spline curve, and the algorithm is implemented in the sensors. It takes into account four position readings for each trackside to calculate six control points. A schematic diagram of the data used to calculate the B-Spline control points is shown in Figure 2.

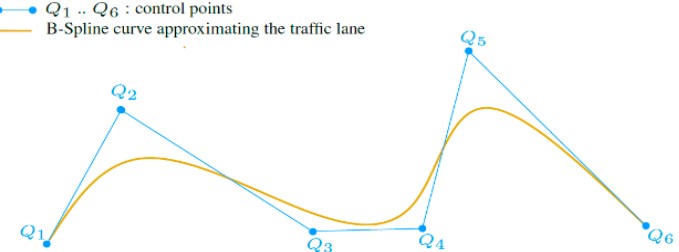

**Figure 2.** Graphical representation of B-Spline curve associated to the control points.

The UMRR radar sensor generates a list of detected targets within its field of view. Each target entry includes information such as:

- **Timestamp:** The time at which the measurement was taken, typically expressed in milliseconds or seconds.
- **Number of objects:** The count of objects detected by the radar sensor during the measurement.
- **Cycle duration:** The duration of a measurement cycle, typically 55 ms for this type of radar.
- **X–Y points:** The coordinates of the object's position (X and Y), typically measured in meters, indicating its longitudinal and transverse position relative to the reference point centered on the sensor.
- **Width:** The width of the detected object, typically measured in meters.
- **Length:** The length of the detected object, typically measured in meters.
- **Speed:** The absolute speed of the detected object, typically measured in meters per second.
- **Heading:** The heading or direction of the object, often measured in degrees.
- **Object ID:** An identifier assigned to the detected object, allowing for tracking and reference.

The binary data flow is converted into ASCII files using an internally developed Dynamic Link Library (DLL). This DLL ensures synchronization of timestamps with the computer internal clock with millisecond precision. Additionally, it filters out irrelevant data from raw recordings by removing empty measurement records.

## 3. Description of the HADER Standard Cinemometer

The HADER standard cinemometer, used as a reference, is based on a Doppler CW radar [13] operating at a frequency of 24.125 GHz. It is equipped with a fixed parabolic antenna that emits a narrow-beam signal with a beamwidth of 1.75° and a maximum sidelobe level of −20 dB, ensuring high directional measurement accuracy. The system is designed to be installed at a 25° angle relative to the road axis. The positioning angle of the

antenna must be determined with an uncertainty of less than 0.1° to maintain measurement reliability. The radar measures vehicle speeds ranging from 30 km/h to 300 km/h, with each measurement being time-stamped to ensure traceability. The antenna emission power is 1 mW, and the radar samples 16 cycles per measurement, corresponding to an equivalent displacement length of 0.11 m.

The reference system undergoes a calibration check every two years. The calibration process consists of two measuring steps. The first step involves measuring the frequency emitted by the cinemometer. The calibration is performed by using a frequency meter controlled by a signal from a rubidium frequency standard, combined with a horn antenna. The second step involves measuring the speeds displayed by the cinemometer against a pre-calibrated Doppler shift simulator. The simulated speeds are computed from the emission frequency of the cinemometer and the 25° angle $\alpha$ related to the radar positioning. These speeds are also influenced by the simulated Doppler frequencies. Its value is given by [14]:

$$V = \frac{cF_d}{2\cos(\alpha)F_0},$$ (1)

with

- $V$ simulated speed in m/s.
- $c$ propagation constant in air equal to 299,702,547 m/s.
- $F_d$ frequency simulating the Doppler effect in Hz.
- $F_0$ frequency emitted by the cinemometer in Hz.
- $\alpha$ positioning angle of cinemometer in rad.

## 4. The Calibration Method

### 4.1. Principle

The calibration of the radar sensors by comparison with the reference cinemometer involves simultaneously measuring the vehicle speeds on the road with the radar sensor and the reference cinemometer.

The initial step involves setting up the HADER system, which includes measuring its angular position with respect to the road and synchronizing its control computer timebase with the reference clock.

The HADER system provides a single speed value for each target it detects, associated with a timestamp. In contrast, the radar sensors provide a list of speed and position values for each target, each also associated with timestamps. The comparison process involves carefully selecting the most appropriate speed value provided by the radar. This chosen value is then matched with the corresponding speed reading from the reference cinemometer for the observed target. In the context of this paper, this crucial operation, referred to as 'pairing', is employed to establish a precise correspondence between the radar sensor speed measurements and those recorded by the reference system for a common target.

The pairing process consists of the following steps:

- First, for each measurement of the HADER cinemometer, the associated targets identified by the radar sensors being calibrated are determined. The recorded data of the targets encompasses the timestamps, the speed readings, and the real-time tracking positions. These data are interpolated concerning the detection position on the longitudinal axis of the road to improve matching accuracy between the radar and HADER systems. The other objects are subsequently filtered out by the algorithm.
- The second step involves adjusting the measured reference speed $V^{meas}$ based on the effective alignment angle $\alpha^{eff}$ of the HADER platform with respect to the road axis. This angle is measured using a theodolite. The HADER speed correction $V^{eff}$ is given by:

$$V^{eff} = V^{meas} \frac{\cos(\alpha)}{\cos(\alpha^{eff})},$$

(2)

- A time offset between the speed cinemometers is calculated using a reference measurement that corresponds to the maximum speed recorded by both systems at the violation position. This offset is compensated by adjusting the HADER timestamps. Typically, offset corrections are below a few hundred milliseconds. Typically, offset corrections are below a few hundred milliseconds.
- Finally, the radar measurements are matched with the corrected reference measurements according to their timestamps. To prevent pairing confusion among objects that are very close when passing the violation position, a temporal criterion is applied during the speed matching process. It is based on the minimum time offset between the consecutive HADER measurements passing through the infringement zone.

The speed values provided by the reference cinemometer are corrected based on the system calibration. The HADER calibration method involves simulating a Doppler shift using a Doppler target simulator operating at 24 GHz [15]. This approach ensures the accuracy of the speed measurements from the reference cinemometer. It aligns these measurements with the calibrated values obtained through the Doppler shift simulator at the specified frequency.

*4.2. Measurement Conditions*

The calibration of the radar sensors is conducted at an outdoor traffic site by comparing the measurements with those from the reference system.

As shown in Figure 3, the radar sensors are mounted on a rotating platform with adjustable bases for each sensor to facilitate precise positioning. The platform is attached to a metal mast structure, positioned 2.4 m above the ground. The structure must be installed parallel to the traffic lane, ensuring that the sensor's pointing zone is clear and encompasses the HADER detection zone. For the calibration tests, the sensor support is typically oriented at an angle of 20° to the traffic lane. At this alignment, the radar's dead zone is estimated to be less than 3 m laterally (perpendicular to the road axis) and 10 m longitudinally (parallel to the road axis), based on the radar mast's location. Therefore, the mast structure should be positioned at least 3 m from the edge of the road.

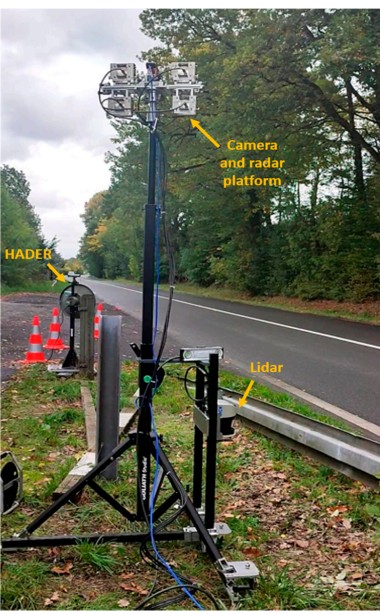

**Figure 3.** Visual representation of the measurement campaign setup.

The reference cinemometer is oriented relative to the traffic lane to establish an angle between the road edge and the radar beam close to the nominal angle of 25°. The actual angle is determined by measuring two aligned points along the beam with a yardstick and a spotting scope mounted on the reference cinemometer, which is aligned with the axis of its emission beam. The road axis is calculated from trackside position measurements taken using a Leica TS10 total station.

### 4.3. Test Locations and Dates

The calibration of radar sensors is conducted through several tests across various types of traffic sites, including departmental roads, highways, and high-speed circuits. Table 1 provides essential information about these tests.

**Table 1.** Calibration tests of radar sensors on different types of traffic site.

|  | Traffic Site Type | Date | GPS Location | Speed Range (km/h) | Paired Vehicles Number |
|---|---|---|---|---|---|
| Test 1 | Highway road | 17 May 2021 | 48°35′57.1″ N 2°03′28.6″ E | 80–140 | 128 |
| Test 2 | Departmental road | 21 May 2021 | 48°41′47.9″ N 2°13′39.2″ E | 50–90 | 63 |
| Test 3 | Departmental road | 20 October 2021 | 48°41′47.9″ N 2°13′39.2″ E | 50–90 | 123 |
| Test 4 | High-speed circuit | 28 January 2021 | 49°08′26.7″ N 2°36′08.4″ E | 90–220 | 94 |
| Test 5 | High-speed circuit | 25 March 2022 | 49°08′26.7″ N 2°36′08.4″ E | 90–220 | 72 |

### 4.4. Data Analysis

#### 4.4.1. Residual Timing Error

To evaluate the average time offset between the radar systems and the reference system, we conduct a post-processing analysis to identify the highest speed recorded by both systems at the violation line. The resulting difference reflects the gap between the timestamps of the two systems. We then adjust the HADER system timestamps to account for this delay. However, a residual random timing error may occur due to this operation, depending on the radar sensor's sampling period.

Figure 4 illustrates the time error variations between HADER and the radar sensor during a 10 min test run on a two-lane road in the suburbs of Paris. The residual standard deviation of the time error between the two systems is 33 ms. This value is less than the radar sensor's sampling period of 55 ms. This observation indicates that the pairing quality is adequate for comparing measured speeds. However, it introduces variability in the speed difference measured by the reference cinemometer and the radar sensor, which is evaluated as part of the type A uncertainty contribution.

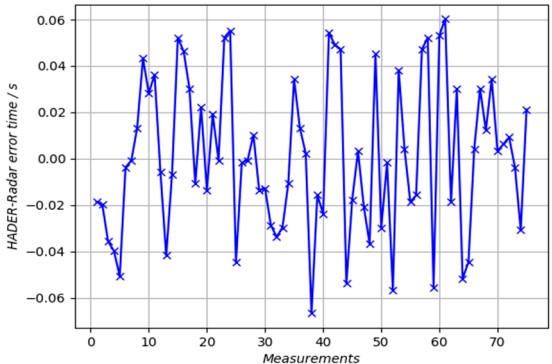

**Figure 4.** Error time variations between HADER and radar sensor for 10 min test on departmental road.

### 4.4.2. Residual Position Error

The radar sensor integrates the detected objects within its view field along the B-Spline curve, which approximates the traffic lane. The B-Spline curve is calculated using a dedicated algorithm based on road measurements. An example of the X–Y trajectories of objects provided by the radar sensor on a road traffic lane is depicted in Figure 5.

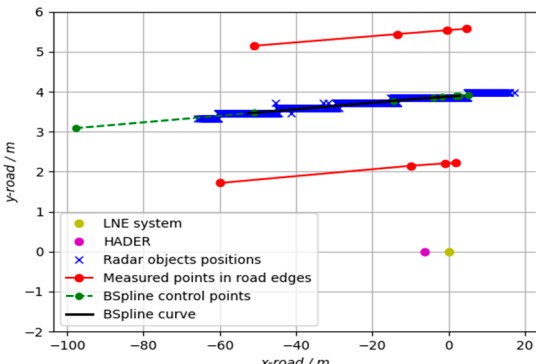

**Figure 5.** Objects positions recorded by radar sensor along a B-Spline curve.

The speed and position data of vehicles are interpolated for analysis to the longitudinal position of the violation line. This interpolation process introduces a positional error along the transverse axis, relative to the violation position, which is also influenced by the spatial resolution of the radar sensor.

Figure 6 illustrates the transverse positional error of objects with respect to the violation line. This error is negligible compared to the estimated lane width of 2.4 m.

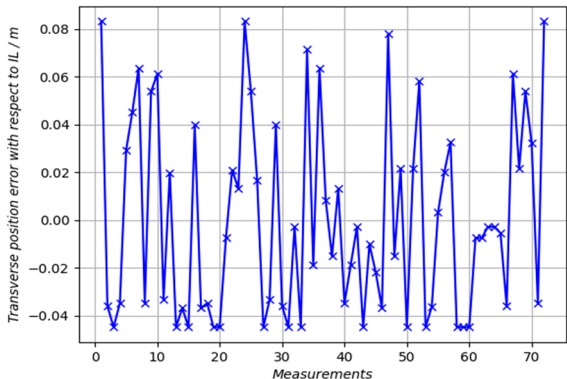

**Figure 6.** Transverse position error of radar objects with respect to the violation line.

### 4.4.3. Pairing Results

The HADER–Radar pairing for each sensor allows us to define the mean $\mu$ and the standard deviation $s$ of the difference between HADER and the radar for the measurement points nearest to the violation line. Table 2 presents an example of this statistical information, showing the variations observed in three trials conducted on 20 October 2021, along a two-lane road in the suburbs of Paris. These trials included speeds ranging from 50 to 90 km/h.

Figure 7 presents a comparison of speed measurements obtained from HADER and radar sensors within the first 10 min measurement session, including the mean relative error and the associated standard deviation. This comparison reveals that the reference and radar systems exhibit a strong correlation in their speed measurements.

**Table 2.** Mean ($\mu$) and standard deviation ($s$) of relative speed difference between HADER and radar sensors under diverse test conditions.

|  |  | Test 10 min | Test 20 min | Test 30 min |
|---|---|---|---|---|
| **HADER–Radar 1** | $\mu/\%$ | 0.15 | 0.13 | 0.18 |
|  | $s/\%$ | 0.40 | 0.39 | 0.47 |
| **HADER–Radar 2** | $\mu/\%$ | $-0.05$ | $-0.09$ | 0.03 |
|  | $s/\%$ | 0.33 | 0.42 | 0.49 |
| **HADER–Radar 3** | $\mu/\%$ | $-0.21$ | $-0.23$ | $-0.27$ |
|  | $s/\%$ | 0.39 | 0.43 | 0.49 |
| **HADER–Radar 4** | $\mu/\%$ | $-0.01$ | $-0.02$ | $-0.08$ |
|  | $s/\%$ | 0.43 | 0.43 | 0.52 |

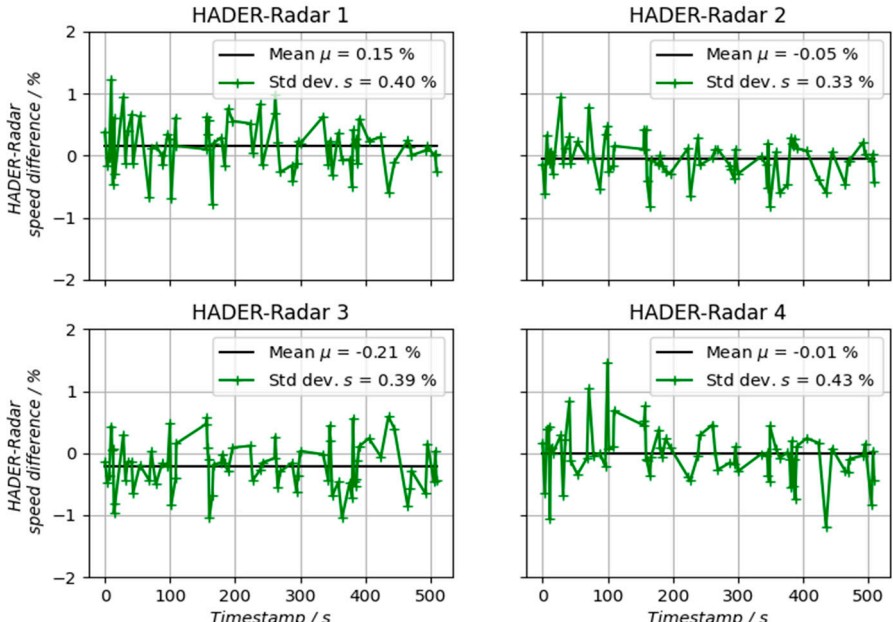

**Figure 7.** Speed difference of HADER–Radar pairing results with indication of mean relative error and associated standard deviation.

*4.5. Uncertainty Analysis*

The uncertainty analysis below provides a method to calculate the uncertainty associated with a single speed value generated by a radar sensor, providing an evaluation of type A uncertainty based on numerous individual measurements (dozens per session over four sessions).

- **A1—Repeatabilty of measurement deviations:** This uncertainty component is assessed by calculating the standard deviation of measurement differences between the HADER system and radar sensors. This evaluation quantifies the variability in results obtained from measurements repeated under identical conditions, offering insight into the consistency of radar measurements with the reference system.
- **B1—Calibration uncertainty of the standard system:** The calibration uncertainty of the reference cinemometer is assessed through laboratory calibration and is periodically verified through comparisons with an external laboratory. The value of this uncertainty component depends directly on the calibrated speed value and is provided in the calibration certificate issued by the laboratory.

- **B2—Standard drift:** This aspect of calibration uncertainty accounts for any drifts in the properties of the HADER reference system over time. It is crucial to monitor and make adjustments to any changes in the reference system's performance. This component depends on the speed measured by the HADER system.
- **B3—HADER temperature variations (10 °C to 30 °C):** The temperature variations within the range of 10–30 °C can introduce uncertainty. Temperature fluctuations may affect the transmitted frequency of the HADER system and, consequently, the calibration accuracy. This component is evaluated as 0.03% within the temperature range of 10–30 °C.
- **B4—Radar temperature variations (10 °C to 30 °C):** The temperature fluctuations within the range of 10–30 °C can introduce uncertainty, potentially impacting radar performance and, subsequently, calibration accuracy. This component is evaluated as 0.03% within the range of 10–30 °C.
- **B5—HADER display resolution:** The resolution of the measurement display in km/h assesses the reference system's ability to present measurements with precision. This parameter ensures that measurements are not only accurate but also presented in a user-friendly format. The resolution is 0.1 km/h relative to the measured speed.
- **B6—Radar display resolution:** Similarly, this evaluation assesses the radar system's measurement display resolution in m/s, emphasizing the importance of an accurate and user-friendly presentation. The resolution is fixed at 0.1 m/s relative to the measured speed.
- **B7—HADER angular positioning:** The accurate angular positioning within the reference system is critical. Misalignment with the effective 25° angle of the HADER orientation can lead to variations in the speed reference measurement. In this study, we account for a positioning uncertainty of 0.1° around the 25° angle.
- **B8—Radar angular positioning:** Like HADER, the angular positioning of the radar platform introduces uncertainty. An angular deviation of 0.1° is considered in relation to the radar's installation angle, set at 20° relative to the road axis.

These components can be classified into two broad categories: those that depend on the measured speed (e.g., standard drift and systematic error) and those that remain constant or independent of the measured speed (e.g., display resolution, calibration, and angular positioning uncertainties). This categorization aids in comprehending the impact of these sources of uncertainty in various measurement scenarios.

Table 3 includes laws, dividing factors, sensitivity coefficients, and assessed values for each uncertainty component, categorized as either variable or constant.

**Table 3.** Table of uncertainty components.

| Component | Law | Dividing Factor | Sensitivity Coefficient | Type |
|---|---|---|---|---|
| A1 | Normal | 1 | 1 | Variable |
| B1 | - | 1 | 1 | Variable |
| B2 | Rectangular | 1 | 1 | Variable |
| B3 | Arc sine | $\sqrt{2}$ | 1 | Constant |
| B4 | Arc sine | $\sqrt{2}$ | 1 | Constant |
| B5 | Rectangular | $\sqrt{3}$ | 1 | Constant |
| B6 | Rectangular | $\sqrt{3}$ | 1 | Constant |
| B7 | Rectangular | $\sqrt{3}$ | 1 | Constant |
| B8 | Rectangular | $\sqrt{3}$ | 1 | Constant |

The overall uncertainty of individual speed values measured by the radar sensors is calculated by combining the individual uncertainties from all components quadratically, ensuring a comprehensive evaluation of radar calibration against the reference system.

Table 4 presents the relative uncertainty values calculated for various speed ranges. The overall uncertainty, expressed as expanded uncertainty with a coverage factor of 2, is 1.3% at 50 km/h, 1.1% at 70 km/h, 1.0% at 90 km/h, 1.1% at 180 km/h, and 1.1% at 200 km/h.

**Table 4.** Results of radar uncertainty measurements for medium-speed and high-speed ranges.

| Component | Medium-Speed Tests | | | High-Speed Tests | |
|---|---|---|---|---|---|
| | 50 km/h | 70 km/h | 90 km/h | 180 km/h | 200 km/h |
| A1/% | 0.43 | 0.43 | 0.43 | 0.51 | 0.51 |
| B1/% | 0.06 | 0.04 | 0.03 | 0.02 | 0.02 |
| B2/% | 0.12 | 0.00 | 0.02 | 0.02 | 0.01 |
| B3/% | 0.07 | 0.07 | 0.07 | 0.07 | 0.07 |
| B4/% | 0.07 | 0.07 | 0.07 | 0.07 | 0.07 |
| B5/% | 0.12 | 0.08 | 0.06 | 0.03 | 0.03 |
| B6/% | 0.42 | 0.30 | 0.23 | 0.10 | 0.06 |
| B7/% | 0.05 | 0.05 | 0.05 | 0.05 | 0.05 |
| B8/% | 0.04 | 0.04 | 0.04 | 0.04 | 0.04 |
| Expanded Uncertainty/% (coverage factor = 2) | **1.3** | **1.1** | **1.0** | **1.1** | **1.1** |

## 5. Discussion

### 5.1. Results and Discussion

In our study, we conducted tests at both medium-speed ranges (50 km/h, 70 km/h, and 90 km/h) and high-speed ranges (180 km/h and 200 km/h) to assess the uncertainty of the unitary speed values measured by the radar sensors. The uncertainties measured, as shown in Table 4, reveal a high degree of precision in our measurements applicable to individual speed values. We evaluate uncertainty according to the previously mentioned uncertainty budget. At these speeds, the overall uncertainties, presented as expanded uncertainty with a coverage factor of 2, are much lower than the French regulatory standards for speed measurements, which establish a threshold of 3 km/h for speeds below 100 km/h and 3% for speeds exceeding 100 km/h.

Our system combines data from multiple sensors to reduce uncertainty compared to individual measurements. The speed uncertainty is calculated by considering the contributions of each sensor and using a fusion algorithm that takes advantage of the sensor's tracking abilities. This process minimizes uncertainty through the averaging of multiple measurements. The fusion algorithm collects data from different sensors, synchronizes timestamps, removes incomplete or incorrect data, applies interpolation, and calculates a reference speed with an associated uncertainty. By leveraging the redundancy of the sensors, the system achieves improved accuracy and reduced uncertainty.

### 5.2. Validation Method

The validation of our calibration method involves comparing the radar results with measurements obtained from a calibrated lidar cinemometer, which has undergone precise rotational and distance calibration. Notably, the calibration method for the lidar system operates independently of the radar sensors used in the HADER reference cinemometer.

The key indicator for this validation is the normalized error $E_N$ between radar and lidar measurements, calculated as follows [16]:

$$E_N = \frac{|V_{radar} - V_{lidar}|}{\sqrt{U_{radar}^2 + U_{lidar}^2}},$$ (3)

with

- $V_{radar}$ the radar speed measurement.
- $V_{lidar}$ the lidar speed measurement.
- $U_{radar}$ the expanded uncertainty associated to the radar measurement.
- $U_{lidar}$ the expanded uncertainty associated to the lidar measurement.

In this comparative study, we observe a maximum absolute uncertainty of 1.3% for the radar measurements, while the lidar measurements exhibit an uncertainty of 0.72%, as determined by the lidar calibration method.

Figure 8 illustrates the results of the normalized error for measurements obtained from four radar sensors compared to those from the lidar sensor during trials conducted on a departmental road.

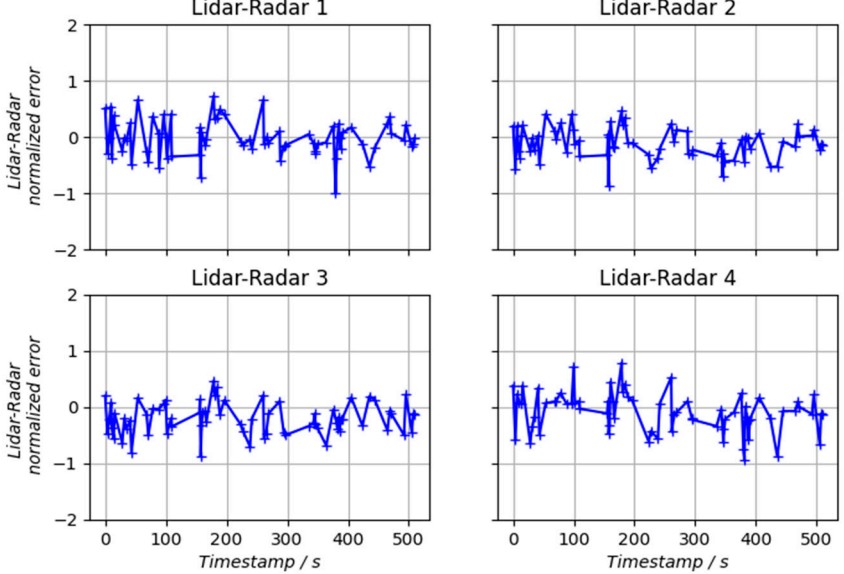

**Figure 8.** Normalized error between radar and lidar measurements for tests performed on a departmental road.

The normalized error does not exceed 1, indicating that the radar and lidar measurements are in agreement. The values obtained from these sensors are closely aligned, showing minimal variation compared to the associated measurement uncertainties.

*5.3. Method Limitations*

The method described for calibrating the radar sensors by comparing them with the HADER reference cinemometer has several limitations that can impact the accuracy and reliability of the calibration process. These limitations include:

- **Speed range limitation:** One significant limitation is the inherent speed range that both the radar sensor and the reference system can accurately measure. If the measurement range does not cover the full spectrum of speeds encountered on the road, gaps in the calibration data may occur. This can lead to inaccuracies, particularly for vehicles traveling outside the effective range of the sensors.

- **Synchronization impact:** Accurate calibration relies on precise timestamp synchronization between the radar sensor and the HADER system. Any timing discrepancies between the two systems can introduce errors in the pairing process, making it challenging to match radar measurements with corresponding reference measurements accurately.
- **Detection errors:** The radar sensors may experience detection errors, resulting in missed targets. If a vehicle is detected by the HADER system but not by the radar sensor, there will be no corresponding data for calibration, leading to an incomplete dataset.
- **Masking effects:** The reference cinemometer can suffer from masking effects when multiple vehicles are close together or moving in parallel. This may complicate the identification of which HADER measurement corresponds to a particular radar measurement, introducing ambiguity and errors in the pairing process.

Collectively, these limitations affect the accuracy and reliability of the calibration method. To mitigate these issues, it is crucial to carefully select calibration data, ensure proper synchronization, and account for detection errors and masking effects during the pairing process. To further address these limitations, several measures can be implemented. Using a high-accuracy lidar as an additional reference can resolve ambiguities caused by multiple vehicles or complex traffic scenarios, improving radar-lidar pairing. Laboratory-based calibration with a target simulator ensures traceability to the International System of Units (SI) by simulating a wide range of controlled velocities and precise timing, which addresses speed range and synchronization issues. Additionally, statistical models and data filtering techniques can correct detection errors, manage incomplete data, and improve the overall accuracy of the calibration process.

*5.4. Method Advantages*

This calibration approach offers several notable advantages, making it a valuable technique for real-world calibration endeavors:

- **Calibration in realistic outdoor conditions:** This method stands out by enabling calibration under authentic operational conditions. By comparing radar sensor measurements with those from the reference cinemometer in the field, it ensures that the calibration aligns with practical scenarios encountered on the road.
- **Comprehensive uncertainty consideration:** Another significant strength lies in its ability to thoroughly account for various sources of uncertainty inherent in real-world settings. By precisely considering these uncertainty components, the calibrated sensor delivers accurate measurements, even in environments influenced by diverse factors.
- **Adaptability to diverse speed measurement devices:** The method is also notable for its adaptability to a wide range of speed sensors. It is not limited to specific types of radar sensors, which grants it versatility for use with various speed measurement technologies. Whether utilizing Doppler radar, FMCW radar, or other speed sensors, this calibration method proves effective.

## 6. Conclusions

In this study, we developed and implemented an on-site calibration method for speed sensors, integrating a tracking function through comparison with an SI-traceable reference cinemometer. This innovative approach enables a comprehensive assessment of the uncertainties associated with on-site calibration, encompassing various speed values measured by individual cinemometers equipped with a tracking function.

One of the main contributions of this research is the provision of a complete uncertainty balance for an on-site calibration method that reflects real-world use cases for

cinemometers with tracking capabilities. This method is adaptable to various sensor technologies, including video, radar, and lidar, as long as the input data include position, speed, and timestamps from multiple locations.

The findings of this research pave the way for future practical applications, including the integration of uncertainty components into the calculation of the system's measured speed. By combining multiple speed values obtained from different sensors, we can reduce the final uncertainty of the reported speed, thereby enhancing the accuracy of speed measurements across diverse contexts.

**Author Contributions:** Conceptualization, S.B.-H. and J.-M.L.; methodology, S.B.-H. and J.-M.L.; post-processing software, S.B.-H., J.-M.L. and D.R.; validation, S.B.-H., J.-M.L. and D.R.; formal analysis, S.B.-H.; investigation, S.B.-H. and J.-M.L.; resources, S.B.-H., J.-M.L., J.D., P.B. and D.R.; data curation, S.B.-H., J.-M.L., J.D., P.B. and D.R.; writing—original draft preparation, S.B.-H. and J.-M.L.; writing—review and editing, S.B.-H. and J.-M.L.; visualization, S.B.-H. and J.-M.L.; supervision, J.-M.L.; project administration, J.-M.L. All authors have read and agreed to the published version of the manuscript.

**Funding:** This research was conducted as a support contract for the French Home Office, Automated Control Direction.

**Data Availability Statement:** Data are contained within the article.

**Conflicts of Interest:** The authors declare no conflicts of interest.

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
