# Peer review of "Calibration of Target-Tracking MIMO Radar Sensors by Comparison with a Narrow-Beam CW Doppler Reference"

_2673-8244, doi:10.3390/metrology5010014_

Round 1
Reviewer 1 Report
Comments and Suggestions for Authors
The authors have presented a novel method for calibrating radar sensors. The contribution of this work is significant. There are several minor comments as follows:
1. In Introduction, further discussion about the related works should be added. The authots should provide the pros and cons of the published works to demonstrate the advantages of the proposed method.
2. The comparison with the conventional methods is also important.
3. The method limitations are thoroughly discussed in Section 5.3. Could the authors proposed the solutions?
Author Response
Comments 1: In Introduction, further discussion about the related works should be added. The authors should provide the pros and cons of the published works to demonstrate the advantages of the proposed method.
Response 1: Agree. We have, accordingly, revised the Introduction section to include a more detailed discussion about related works and their respective advantages and limitations. Specifically, we have added references to existing methods, including pulsed CW radar systems, piezo sensors, inductive loops, and GPS-based speed measurement systems, to provide a broader context and better emphasize the relevance of the proposed method.
The changes can be found on Introduction section, page 1, of the revised manuscript.
Updated text in the manuscript:
“Speed control for law enforcement application has evolved significantly with the development of modern MIMO radar sensors and lidar sensors. New features such as measurement on curves, simultaneous measurement of several targets in multiple lanes, and speed gradient measurement are now common, and their validation requires the development of new reference systems. Existing solutions are based on reference cinemometers using a pulsed Continuous-Wave (CW) radar with a center frequency of 24.125 GHz and a fixed incidence angle of 25° [1]. This type of solution provides reliable speed measurements but suffers from several limitations, such as limited multi-lane detection capability, installation restricted to straight-line configurations, and the ability to detect only a single target at a time. Other approaches found in the literature include piezoelectric sensors and inductive loops installed in the road [2]. These solutions offer good measurement accuracy but require a dedicated test circuit and do not allow for the relocation of the reference system to different sites. This makes it challenging to test cinemometers on single-lane roads, multi-lane traffic scenarios, or curved roads.
Finally, the reference system for law enforcement cinemometer validation can also be a speed measurement system integrated into a test vehicle. GPS-based solutions have been tested with various cinemometers in [3]. These solutions enable the verification of cinemometers, provided there is accurate synchronization between the device under test and the GPS-based reference speed sensor.
Consequently, a new metrological reference was developed at Laboratoire National de Métrologie et d’Essais (LNE) to provide a solution that adapts to these advancements and addresses the shortcomings of existing methods. This new reference allows on site testing of cinemometer systems with improved detection capabilities and new functionality compared to the previous one which is based on a single CW radar.”
Comments 2: The comparison with the conventional methods is also important.
Response 2: A synthesis of conventional methods is presented in the Introduction, with a focus on the pulsed CW radar system used a reference for the present study. Comparison with existing methods is mainly based on practical aspects as the paper presents the results of a given calibration method of our sensor rather than a review of existing methods. Criteria resulting in the choice of our method are including the flexibility of the method, the accuracy of the reference and the budgetary advantages of relying on an existing system.
Comments 3: The method limitations are thoroughly discussed in Section 5.3. Could the authors proposed the solutions?
Response 3: Agree. We have, accordingly, revised Section 5.3 to propose solutions that address the method limitations and emphasize this point. We have added suggestions such as using a high-accuracy lidar system to resolve ambiguities caused by masking effects, laboratory-based calibration with a target simulator to address speed range and synchronization issues, and statistical techniques to improve the accuracy of the calibration process by reducing errors and managing incomplete data.
These additions can be found in Section 5.3, page 12.
Updated text in the manuscript:
“Collectively, these limitations affect the accuracy and reliability of the calibration method. To mitigate these issues, it is crucial to carefully select calibration data, ensure proper synchronization, and account for detection errors and masking effects during the pairing process. To further address these limitations, several measures can be implemented. Using a high-accuracy lidar as an additional reference can resolve ambiguities caused by multiple vehicles or complex traffic scenarios, improving radar-lidar pairing. Laboratory-based calibration with a target simulator ensures traceability to the International System of Units (SI) by simulating a wide range of controlled velocities and precise timing, which addresses speed range and synchronization issues. Additionally, statistical models and data filtering techniques can correct detection errors, manage incomplete data, and improve the overall accuracy of the calibration process.”
Reviewer 2 Report
Comments and Suggestions for Authors
This paper introduces a method for calibrating radar sensors in multi-sensor cinemometer system using a reference cinemometer based on continuous wave Doppler radar, which has certain innovation and practical value. In general, the mentality of the paper is clear and language organization is fluent, but some technical content needs to be supplemented. Revision suggestions are listed in the following :
(1) Why does the HADER standard cinemometer system have to form a 25 ° angle with the road, and the 77GHz millimeter-wave radar has to form a 20 ° angle with the road? Is there a theoretical basis for selecting the working angles of the two radars? Please provide a detailed explanation.
(2) The operation of HADER system and millimeter-wave radar system is completely asynchronous. The radar system uses signal processing methods to obtain target velocity and angle information, which consumes processing time. Does the timestamp positioning method used in the article have the problem of drift over time? The asynchronous operation between sensors is uncontrollable. Please provide a detailed introduction and quantitative analysis on how to calibrate the speed of each sensor based on timestamps.
(3) There is relatively little introduction to the technical content of the 24GHz Doppler radar as a HADER standard speedometer in the article, whether it is a standardized series product or a specially customized product. It is necessary to supplement the selection parameters of the radar, and provide more detailed technical content such as ranging accuracy, speed measurement accuracy, angle measurement accuracy, and update rate of the radar.
(4) The text provides only textual descriptions of the calibration process, and it is necessary to supplement the mechanical structure installation diagram of the HADER standard speedometer and speed radar calibration process, clearly providing a three-dimensional design scheme for the test scenario, radar installation position, height, and angle.
(5) The article has relatively few references, it is recommended to supplement some of them.
Author Response
Comments 1: Why does the HADER standard cinemometer system have to form a 25 ° angle with the road, and the 77GHz millimeter-wave radar has to form a 20 ° angle with the road? Is there a theoretical basis for selecting the working angles of the two radars? Please provide a detailed explanation.
Response 1: The HADER cinemometer system is designed to be installed at a 25° angle to the road as part of its technical specifications. This angle is required for the system to work properly and provide accurate measurements. The 77GHz millimeter-wave radar is set at a 20° angle to balance how far it can see and how much of the road it can cover. This angle is especially helpful on multi-lane roads and curves, where it needs to track several vehicles at the same time. Tests have shown that this angle offers a good balance between covering a wide area and maintaining reliable tracking of moving vehicles in different traffic conditions.
Comments 2: The operation of HADER system and millimeter-wave radar system is completely asynchronous. The radar system uses signal processing methods to obtain target velocity and angle information, which consumes processing time. Does the timestamp positioning method used in the article have the problem of drift over time? The asynchronous operation between sensors is uncontrollable. Please provide a detailed introduction and quantitative analysis on how to calibrate the speed of each sensor based on timestamps.
Response 2: Agree. Although the HADER system and the 77 GHz millimeter-wave radar operate asynchronously, which can lead to timestamp drift due to differences in processing times, the proposed calibration method effectively addresses this issue.
First, the timestamps from both the HADER and radar systems are synchronized using the MasterClock GR1000 reference clock, reducing the initial timing discrepancies. The radar data (timestamps, speeds, and positions) are then interpolated based on the detection position along the longitudinal axis of the road to improve the matching accuracy between the two systems. Simultaneously, the speed measured by the HADER system is corrected based on the alignment angle between the HADER platform and the road axis, enhancing speed measurement accuracy.
A time offset between the HADER and radar systems is calculated using a reference measurement corresponding to the maximum speed recorded by both systems at the point of infraction. This time offset, typically less than a few hundred milliseconds, is compensated for by adjusting the timestamps of the HADER system. The radar data is then matched with the corrected HADER measurements based on the adjusted timestamps, by applying a temporal criterion to avoid matching errors for nearby objects.
This process ensures reliable pairing between radar and HADER data, even with asynchronous operation. The results presented in Section 4.4.1 show that the temporal error between HADER and radar after pairing has a standard deviation of 33 ms, which is below the radar sampling period of 55 ms, as illustrated in Figure 4. This indicates that the pairing quality is high and the impact of asynchronism between the two systems is negligible, even at high speeds.
Comments 3: There is relatively little introduction to the technical content of the 24 GHz Doppler radar as a HADER standard speedometer in the article, whether it is a standardized series product or a specially customized product. It is necessary to supplement the selection parameters of the radar, and provide more detailed technical content such as ranging accuracy, speed measurement accuracy, angle measurement accuracy, and update rate of the radar.
Response 3: Additional technical specifications of HADER system have been provided in Section 3 on page 3 to address this point.
Updated text in the manuscript:
“The HADER standard cinemometer, used as a reference, is based on a Doppler CW radar [7] operating at a frequency of 24.125 GHz. It is equipped with a fixed parabolic antenna that emits a narrow-beam signal with a beamwidth of 1.75° and a maximum sidelobe level of -20 dB, ensuring high directional measurement accuracy. The system is designed to be installed at a 25° angle relative to the road axis. The positioning angle of the antenna must be determined with an uncertainty of less than 0.1° to maintain measurement reliability. The radar measures vehicle speeds ranging from 30 km/h to 300 km/h, with each measurement being time-stamped to ensure traceability. The antenna emission power is 1 mW, and the radar samples 16 cycles per measurement, corresponding to an equivalent displacement length of 0.11 meters.”
Comments 4: The text provides only textual descriptions of the calibration process, and it is necessary to supplement the mechanical structure installation diagram of the HADER standard speedometer and speed radar calibration process, clearly providing a three-dimensional design scheme for the test scenario, radar installation position, height, and angle.
Response 4: A visual representation of the measurement campaign setup has been added in Section 4.2, page 5.
Updated text in the manuscript:
“The calibration of the radar sensors is conducted at an outdoor traffic site by comparing the measurements with those from the reference system.
As shown in Figure 3, the radar sensors are mounted on a rotating platform with adjustable bases for each sensor to facilitate precise positioning. The platform is attached to a metal mast structure, positioned 2.4 m above the ground. The structure must be installed parallel to the traffic lane, ensuring that the sensor’s pointing zone is clear and encompasses the HADER detection zone. For the calibration tests, the sensor support is typically oriented at an angle of 20° to the traffic lane. At this alignment, the radar's dead zone is estimated to be less than 3 m laterally (perpendicular to the road axis) and 10 m longitudinally (parallel to the road axis), based on the radar mast's location. Therefore, the mast structure should be positioned at least 3 m from the edge of the road.”
Figure 1. Visual representation of the measurement campaign setup.
Comments 5: The article has relatively few references, it is recommended to supplement some of them.
Response 5: Additional references have been added to the manuscript.
References to published works have been incorporated into the Introduction section, page 1. Furthermore, a reference for the description of the GMR 1000 time server has been included in Section 2, page 2.
Reviewer 3 Report
Comments and Suggestions for Authors
The paper contains an apparent original method with apparent good results, although some details remain unclear to the reader. Attched the manuscript with some comments:. main issues:
-line 122: you quote c in vacuum but it is in air. In geodesy extensive equations are known to make the correction. Mayby much smaller than the percents you mention here, but it must be considered and mentioned; also in the framework of traceability.
- table 1: sinus has a different meaning in English; use 'sine'
- not sure if the legislation thresholds are the allowed speed exceedings or the allowed measurement uncertainty to determine them. Anyhow please specify both and underline their differece.
- line 336: without giving the full methods you still must mention how it works and how sensitive you method is for this.
- check space between quantity and units, especially for temperature
For notation of tables and figure axes: see section 5.4.1 of SI brochure : https://www.bipm.org/en/publications/si-brochure

Author Response
Comments 1: line 122: you quote c in vacuum but it is in air. In geodesy extensive equations are known to make the correction. Mayby much smaller than the percents you mention here, but it must be considered and mentioned; also in the framework of traceability.
Response 1: Agree. We made an error by quoting the speed of light in a vacuum rather than the correct value for air. The actual speed of light in air is 299,702,547 m/s, with a refractive index of 1.000273. This value has been corrected in Section 3, page 4, line 139.
Updated text in the manuscript:
"with
- V simulated speed in m/s.
- c propagation constant in air equal to 299 702 547 m/s.
- Fd frequency simulating the Doppler effect in Hz.
- F0 frequency emitted by the cinemometer in Hz.
- α positioning angle of cinemometer in rad. "
Comments 2: table 1: sinus has a different meaning in English; use 'sine'
Response 2: We have corrected the term "sinus" to "sine" in Table 3, page 6, to reflect the proper mathematical terminology in English.
Comments 3: not sure if the legislation thresholds are the allowed speed exceeding or the allowed measurement uncertainty to determine them. Anyhow please specify both and underline their difference.
Response 3: There is no specific requirement in French regulations regarding the uncertainty of a reference cinemometer. The system is used for the approval of new law enforcement cinemometers, so the focus is on performance. The goal is to verify the cinemometer being tested with a sufficient level of confidence, without setting strict limits on the uncertainty of the reference cinemometer.
Comments 4: line 336: without giving the full methods you still must mention how it works and how sensitive you method is for this.
Response 4: The primary objective of this study is to calibrate the radar sensors individually with the HADER system. While the paper does not focus on data fusion, we recognize its role in reducing uncertainty in a multi-sensor system by leveraging redundancy.
To address your comment, we have added a brief description of the fusion algorithm. The algorithm groups data from multiple sensors, applies synchronization, filters out incomplete or aberrant data, performs specific interpolation, and determines a reference speed with an associated uncertainty. This approach enhances the system's overall accuracy and reduces uncertainty compared to individual sensor measurements.
These clarifications have been incorporated into Section 5.1, page 11.
Updated text in the manuscript:
In our study, we conducted tests at both medium-speed ranges (50 km/h, 70 km/h, and 90 km/h) and high-speed ranges (180 km/h and 200 km/h) to assess the uncertainty of the unitary speed values measured by the radar sensors. The uncertainties measured, as shown in Table 4, reveal a high degree of precision in our measurements applicable to individual speed values. We evaluate uncertainty according to the previously mentioned uncertainty budget. At these speeds, the overall uncertainties, presented as expanded uncertainty with a coverage factor of 2, are much lower than the French regulatory standards for speed measurements, which establish a threshold of 3 km/h for speeds below 100 km/h and 3% for speeds exceeding 100 km/h.
Our system combines data from multiple sensors to reduce uncertainty compared to individual measurements. The speed uncertainty is calculated by considering the contributions of each sensor and using a fusion algorithm that takes advantage of the sensor's tracking abilities. This process minimizes uncertainty through the averaging of multiple measurements. The fusion algorithm collects data from different sensors, synchronizes timestamps, removes incomplete or incorrect data, applies interpolation, and calculates a reference speed with an associated uncertainty. By leveraging the redundancy of the sensors, the system achieves improved accuracy and reduced uncertainty.
Comments 5: check space between quantity and units, especially for temperature.
Response 5: We have reviewed and corrected the spacing between quantities and units throughout the text, ensuring consistency with standard formatting conventions (e.g., a space is added between the value and the unit, as per SI guidelines).
Comments 6: For notation of tables and figure axes: see section 5.4.1 of SI brochure : https://www.bipm.org/en/publications/si-brochure
Response 6: Agree. We have reviewed and updated the notation of tables and figure axes to comply with the guidelines provided in Section 5.4.1 of the SI Brochure.
Attached, you will find the responses to your comments written directly on the PDF revision file

Reviewer 4 Report
Comments and Suggestions for Authors
See the attached review document.

Author Response
Comments 1: Please provide the full name of the abbreviation, such as “LNE” and “HADER”.
Response 1: We have clarified the abbreviations as requested. The full name of “LNE” (Laboratoire National de Métrologie et d’Essais) has been added in the Introduction section. Regarding “HADER,” it is a designation given by the manufacturer and does not correspond to an abbreviation.
Comments 2: For this multi-sensor system, how does the number of the acquisition devices (radar sensor, lidar sensor) affect the data error and accuracy? How does that relate to the complexity of traffic (topological B-spline improvement might require for even complicated traffic condition, such as viaduct and three/four-way intersection traffic lane)?
Response 2:
The redundancy provided by multiple acquisition devices, such as radar and lidar sensors, can effectively reduce uncertainty levels by leveraging the combined data. This article focuses specifically on the independent calibration of radar sensors. Following calibration, a dedicated fusion algorithm is applied to combine the data from all sensors, calculate a reference speed, and determine the associated uncertainty.
The radar sensors are already configured using a B-spline algorithm to handle different types of roads, ensuring robust performance even in complex traffic scenarios such as viaducts or intersections. Similarly, the lidar system is designed with flexible configurations to adapt to complex road geometries, enabling accurate measurements in challenging environments.
Round 2
Reviewer 2 Report
Comments and Suggestions for Authors
This paper has been modified in a targeted manner according to the previous review comments, and some missing content has been supplemented with corresponding descriptions. The language organization of the article is concise, the use of data is relatively scientific and reliable, and the use of charts is clear and accurate. Overall, this article has good engineering application value and academic value, and it is recommended to be accepted and published.